Phylogeographic analysis of long-legged bats, Macrophyllum macrophyllum, with notes on roosting behavior and natural history

Garbino Guilherme S.T. guilherme.garbino@ufv.br 1
Semedo Thiago Borges Fernandes 2 3 4
Saldanha Juliane 5
Ferreira Daniela Cristina 5
Rossi Rogerio Vieira 5
da Silva Maria Nazareth Ferreira 6
Lim Burton K. 7
1 Departamento de Biologia Animal, Universidade Federal de Viçosa , Viçosa , Minas Gerais , Brazil
2 CIBIO, Centro de Investigação em Biodiversidade e Recursos Genéticos, Universidade do Porto , Vairão , Portugal
3 Departamento de Biologia, Universidade do Porto , Porto , Portugal
4 Campus de Vairão, BIOPOLIS Program in Genomics, Biodiversity and Land Planning , Vairão , Portugal
5 Instituto de Biociências, Universidade Federal de Mato Grosso , Cuiabá , Mato Grosso , Brazil
6 Coleção de mamíferos, Instituto Nacional de Pesquisas da Amazônia , Manaus , Amazonas , Brazil
7 Royal Ontario Museum , Toronto , Canada
Zhang Lin
Electronic publication date: 2025 May 23
Publication date: 2025
Volume: 13
Electronic Location ID: e19432
Received 2025 Feb 14; Accepted 2025 Apr 16
Copyright: ©2025 Garbino et al.
Copyright year: 2025
Copyright holder: Garbino et al.
License: This is an open access article distributed under the terms of the Creative Commons Attribution License, which permits unrestricted use, distribution, reproduction and adaptation in any medium and for any purpose provided that it is properly attributed. For attribution, the original author(s), title, publication source (PeerJ) and either DOI or URL of the article must be cited.
License URL: https://creativecommons.org/licenses/by/4.0/

Keywords: Phylogeography, Haplogroup, Cytochrome b, Leaf-nosed bat, Cerrado, Pantanal, Phylogeny, Neotropics, Phyllostomidae, Macrophyllini

Funding: The Portuguese Foundation for Science and Technology (Scholarship No. 202210212BD) The American Society of Mammalogists Thiago Borges Fernandes Semedo is supported by a fellowship from the Portuguese Foundation for Science and Technology (Scholarship No. 202210212BD). Guilherme S.T. Garbino received a grant from the American Society of Mammalogists (Oliver Pearson award). The funders had no role in study design, data collection and analysis, decision to publish, or preparation of the manuscript.

==============================
The long-legged bat (Macrophyllum macrophyllum) is widely distributed in the continental Neotropics, but poorly known because it is not commonly caught in mist nets. Available data suggest that this species is closely associated with water where it forages for insect prey. We compiled the first comprehensive molecular dataset assembled for the species, spanning its entire distributional range to investigate if the phylogeography of this monotypic genus is associated with the hydrographic drainage, ecosystem regions, or genetic clustering in Central and South America. To survey under sampled areas, fieldwork was conducted in the Brazilian Pantanal and Cerrado targeting the search for riverine roost sites of Macrophyllum. A literature review was also done to summarize roosting information for the species. New sequences of the mitochondrial cytochrome b gene were generated for tissue samples from Brazil and in museum collections. Phylogenetic trees were constructed using both maximum likelihood and Bayesian inference methods and a haplotype network was used to analyze population structure. Our phylogenetic results identified five geographic lineages of Macrophyllum from (1) the western Cerrado, (2) eastern Cerrado and Pantanal, (3) Guianas, (4) Amazonia, and (5) Central America. However, the haplotype network in conjunction with the genetic clustering identified four populations with the eastern Cerrado and Pantanal grouping with the Guianas and the eastern part of Amazonia. The fieldwork in the Cerrado and Pantanal along with the literature review identified that about half of the roost sites for the long-legged bats were drainage culverts. There is geographic structuring in the mitochondrial data of Macrophyllum with Central America, western Cerrado, Pantanal, Guianas, and eastern Ecuador reciprocally monophyletic and well differentiated populations. However, the under sampled eastern Amazonia is poorly resolved in relation to the other areas. The long-legged bats seem to be relatively adaptable to certain levels of human disturbance and landscape development with man-made drainage culverts commonly used as roosting sites. Increased biodiversity surveys of bats in central Brazil are needed to fill in distributional gaps, such as the lower Amazon River basin, to resolve phylogeographic patterns of Macrophyllum in South America and better understand the potential of cryptic species in this monotypic genus.

Introduction

Knowledge of Neotropical bat diversity has substantially improved over the past two decades (Tsang et al., 2016). As a result of an integrative approach combining genetic and morphological data, recent studies have demonstrated that many widespread species are species complexes (Velazco & Patterson, 2013; Moras et al., 2016; Moras et al., 2024; Novaes et al., 2022; Biganzoli-Rangel et al., 2023; Camacho et al., 2024). In this context, mitochondrial DNA sequence data have enhanced our understanding of bat diversity at both the species and population levels (Martins et al., 2009; Pavan et al., 2011; Garbino, Lim & Tavares, 2020; Silva et al., 2024).

With approximately 230 species, Phyllostomidae is the largest bat family in the Neotropics and has experienced ongoing taxonomic revisions in recent years (Mammal Diversity Database, 2024). Among the lesser-known members of this family is the genus Macrophyllum Gray, 1838, which includes small, long-legged bats (forearm 34–41 mm, weight 6–11 g) characterized by a fully enclosed tail within the large uropatagium that has rows of papillae on its posterior edge (Linares, 1966; Harrison, 1975; Taddei, 1975; Cirranello et al., 2016; Solari et al., 2019). Currently, the genus consists of a single species, Macrophyllum macrophyllum (Schinz, 1821), distributed widely but patchily from southern Mexico to northeastern Argentina (Solari et al., 2019). It is the only phyllostomid bat known to have a trawling behavior, using its long legs and large uropatagium to capture small insects from the water surface (Weinbeer, Kalko & Jung, 2013). The species has a relatively large home range (median 23.9 ha) and a gregarious roosting habit, with colonies typically consisting of 2 to 60 individuals (Solari et al., 2019). Macrophyllum macrophyllum is rarely captured with conventional mist nets, resulting in its natural history being poorly understood, although it is usually found near streams in tropical forests (Thomas, 1928; Handley, 1957; Handley, 1976; Harrison & Pendleton, 1974; Stutz et al., 2004). Additionally, its genetic structure remains largely undocumented, with DNA sequence data available from only a few geographic regions, including the Guianas, western Amazonia, and Central America (Hoffmann, Hoofer & Baker, 2008; Clare et al., 2011).

In this study, we analyze genetic variation in Macrophyllum using the mitochondrial cytochrome b gene to explore its phylogeographic patterns across the distributional range of the genus. Given its frequent association with water bodies for foraging and roosting, our initial hypothesis is that the species’ populations are geographically structured in relation to hydrographic features in the Neotropics. We also test whether major geographic barriers, such as the Andes or the South American dry diagonal, influence the species’ phylogeographic structure. Additionally, we present new natural history observations of the species from central Brazil, in a region where the genus was previously unknown to occur.

Materials & Methods

Specimen sampling and literature review

Bats were captured from their daytime roosts near rivers or creeks at three sites located in the municipalities of Cuiabá in the Cerrado and Santo Antônio do Leverger in the Pantanal, both in the Brazilian state of Mato Grosso (Table 1). Sampling occurred between February 2021 and December 2023. To capture the bats, mist nets were placed around the entrances of their shelters, and as the animals were disturbed by human presence, they flew and became entangled in the nets. Collecting sites are in the Cerrado and Pantanal ecoregions sensu Olson et al. (2001) from elevations below 150 m above sea level. Climate in both areas is classified as “Aw”, or “Tropical with dry winter”, following Köppen’s classification (Alvares et al., 2013). Annual variations in temperature and rainfall are characterized by being markedly seasonal, with a dry season from April to September and a rainy season from September to March.

Table 1 Voucher, tissue, and GenBank accession numbers, sequence length, and localities of Macrophyllum macrophyllum cytochrome b gene sequences analyzed in this study.

Columns to the right represent which population each specimen was assigned based on Bayesian hierarchical clustering (BHC), ecosystem of occurrence, and hydrographic basin. For collection acronyms, see Materials & Methods.

Voucher	Tissue	Accession number	True length (bp)	Country	Locality	Coordinates	BHC	Ecosystem	Hydrographic basin	
IDSM 1348	–	PV037047	721	Brazil	Amazonas, Estação Ecológica Juami-Japurá	02°17′24″S 68°21′32″W	Pop3	Amazonia	Amazon	
IDSM 1349	–	PV037048	767	Brazil	Amazonas, Estação Ecológica Juami-Japurá	02°17′24″S 68°21′32″W	Pop3	Amazonia	Amazon	
INPA 8459	SISJAP-B 785	PV037043	801	Brazil	Amazonas, middle Rio Japurá, right margin of Rio Japurá	01°30′15″S 69°01′53″W	Pop3	Amazonia	Amazon	
INPA 4484	JLP 16741	PV037042	801	Brazil	Amazonas, Miratucú, left margin of Rio Jaú	01°56′S 62°49′W	Pop3	Amazonia	Amazon	
LMUSP	ABX131	PV037044	801	Brazil	Amazonas, Paraná da Eva, close to mouth of Rio Preto da Eva	03°16′30″S 59°05′08″W	Pop3	Amazonia	Amazon	
MZUFV 5089	TS289	PV037054	783	Brazil	Mato Grosso, Comunidade Baía São João, Santo Antônio do Leverger	16°46′12″S 55°33′37″W	Pop2	Pantanal	La Plata	
MZUFV 5170	JS35	PV037049	801	Brazil	Mato Grosso, Comunidade Baía São João, Santo Antônio do Leverger	16°46′12″S 55°33′37″W	Pop2	Pantanal	La Plata	
MZUFV 5172	JS42	PV037052	693	Brazil	Mato Grosso, Comunidade Baía São João, Santo Antônio do Leverger	16°46′12″S 55°33′37″W	Pop2	Pantanal	La Plata	
UFMT 5012	JS36	PV037050	801	Brazil	Mato Grosso, Comunidade Baía São João, Santo Antônio do Leverger	16°46′12″S 55°33′37″W	Pop2	Pantanal	La Plata	
UFMT 5013	JS37	PV037051	801	Brazil	Mato Grosso, Comunidade Baía São João, Santo Antônio do Leverger	16°46′12″S 55°33′37″W	Pop2	Pantanal	La Plata	
UFMT 5014	JS43	PV037040	801	Brazil	Mato Grosso, Comunidade Baía São João, Santo Antônio do Leverger	16°46′12″S 55°33′37″W	Pop2	Pantanal	La Plata	
MZUFV 5467	TS311	PV037034	801	Brazil	Mato Grosso, Ecoville Chapada, Cuiabá	15°11′52″S 55°59′51″W	Pop1	Cerrado	La Plata	
MZUFV 5468	TS312	PV037035	801	Brazil	Mato Grosso, Ecoville Chapada, Cuiabá	15°11′52″S 55°59′51″W	Pop1	Cerrado	La Plata	
UFMT 5015	JS310	PV037036	788	Brazil	Mato Grosso, Ecoville Chapada, Cuiabá	15°11′52″S 55°59′51″W	Pop1	Cerrado	La Plata	
MZUFV 5173	JS282	PV037038	801	Brazil	Mato Grosso, Mimoso, Santo Antônio do Leverger	16°17′17″S 55°40′41″W	Pop2	Pantanal	La Plata	
MZUFV 5174	JS284	PV037040	799	Brazil	Mato Grosso, Mimoso, Santo Antônio do Leverger	16°17′17″S 55°40′41″W	Pop2	Pantanal	La Plata	
UFMT 5016	JS280	PV037037	801	Brazil	Mato Grosso, Mimoso, Santo Antônio do Leverger	16°17′17″S 55°40′41″W	Pop2	Pantanal	La Plata	
UFMT 5017	JS283	PV037039	801	Brazil	Mato Grosso, Mimoso, Santo Antônio do Leverger	16°17′17″S 55°40′41″W	Pop2	Pantanal	La Plata	
UFMT 5018	JS285	PV037041	801	Brazil	Mato Grosso, Mimoso, Santo Antônio do Leverger	16°17′17″S 55°40′41″W	Pop2	Pantanal	La Plata	
UFMG 6819	–	PV037046	801	Brazil	Minas Gerais, Jequitaí	17°14′S 44°26′W	Pop2	Cerrado	São Francisco	
UFPB	PR2020-07	PV037045	801	Brazil	Pará, Vitória do Xingu	03°25′52″S 51°41′29″W	Pop2	Amazonia	Amazon	
ROM 104032	F37130	PV009324	1,140	Ecuador	Napo, Parque Nacional Yasuni, 42 Km S, 1 Km E Pompeya Sur	00°40′48″S 76°25′48″W	Pop3	Amazonia	Amazon	
ROM 104389	F37219	PV009325	1,140	Ecuador	Napo, Parque Nacional Yasuni, 42 Km S, 1 Km E Pompeya Sur	00°40′48″S 76°28′12″W	Pop3	Amazonia	Amazon	
ROM 115718	F51082	PV009333	1,140	Guyana	Potaro-Siparuni, Iwokrama Field Station	04°28′12″N 58°46′48″W	Pop2	Guianas Shield	Northeast South America	
ROM 104684	F38275	PV009328	1,140	Guyana	Potaro-Siparuni, Iwokrama Reserve, 25 km SSW of Kurupukari	04°28′12″N 58°46′48″W	Pop2	Guianas Shield	Northeast South America	
ROM 104698	F38289	PV009329	1,140	Guyana	Potaro-Siparuni, Iwokrama Reserve, 25 km SSW of Kurupukari	04°28′12″N 58°46′48″W	Pop2	Guianas Shield	Northeast South America	
ROM 111632	F44761	PV009332	1,140	Guyana	Potaro-Siparuni, Kabukalli Landing, Iwokrama Forest	04°16′48″N 58°31′12″W	Pop2	Guianas Shield	Northeast South America	
ROM 106563	F38560	PV009330	1,140	Guyana	Upper Takutu-Upper Essequibo, Chodikar River, 55 km SW of Gunn’s Strip	01°22′12″N 58°46′12″W	Pop2	Guianas Shield	Northeast South America	
ROM 106765	F38762	PV009331	1,140	Guyana	Upper Takutu-Upper Essequibo, Chodikar River, 55 km SW of Gunn’s Strip	01°22′12″N 58°46′12″W	Pop2	Guianas Shield	Northeast South America	
ROM 104199	F38026	PV009326	1,140	Panama	Colón, Canal Zone, Gamboa	09°06′00″N 79°42′00″W	Pop4	Central American forests	Pacific	
ROM 104200	F38028	PV009327	1,140	Panama	Colón, Canal Zone, Gamboa	09°06′ 00″N 79°42′00″W	Pop4	Central American forests	Pacific	
ROM 117379	F54700	PV009334	1,140	Suriname	Sipaliwini, Bakhuis, Transect 13	04°39′36″N 57°10′48″W	Pop2	Guianas Shield	Northeast South America	
CMNH 78289	TK19119	FJ155484	1,140	Venezuela	Bolívar, 8 Km S, 5 Km E El Manteco	07°20′N 62°32′W	Pop2	Guianas Shield	Northeast South America	

We reviewed all known published records of the species by searching for the term “Macrophyllum macrophyllum” on Google Scholar and examining the literature cited in the species accounts published by Harrison (1975) and Williams & Genoways (2008). We extracted information on the types of roosts used, colony sizes, and other species inhabiting the same roosts as M. macrophyllum. Roost descriptions that were too vague or general were excluded from our compilation. For instance, Wied (1826) mentioned that the species “spends the day sitting on the rocks and old trunks of the forest” but gave no additional detail. We also ensured that duplicate records were not included. For example, the roosting record reported by Voss et al. (2016) and Velazco et al. (2021) refers to the same observation, and we cited only the earliest reference.

Permits to collect the animals

The capture and handling of specimens complied with the current guidelines for the use of wild mammals in research of the American Society of Mammalogists (Sikes & ASM Animal Care and Use Committee of the American Society of Mammalogists, 2016). In this context, we adhered to the 3R principle—Replacement, Reduction, and Refinement—to optimize ethical and methodological standards (Field et al., 2019). This research was approved by the Animal Care and Use Committee of the Universidade Federal de Viçosa (CEUA-UFV process number 06/2025). Permits for specimen collection were granted by the Instituto Chico Mendes de Conservação (SISBIO permits 76825, 77787) under the Ministry of the Environment, Brazil. The specimens were initially fixed in formalin and preserved in 70% ethanol and are now housed in the Brazilian collections of the Museu de Zoologia João Moojen (MZUFV) in Viçosa and the Coleção de Mamíferos da Universidade Federal de Mato Grosso (UFMT) in Cuiabá. Tissue samples (liver and pectoral muscles) were also collected and preserved in 90% ethanol.

DNA extraction, amplification, and sequencing

Fresh tissue of a total of 32 specimens preserved in ethanol and liquid nitrogen were included in the molecular sampling (Table 1). Voucher specimens are deposited in the following collections: Carnegie Museum of Natural History (CMNH), USA; Centro de Coleções Taxonômicas da Universidade Federal de Minas Gerais (UFMG), Brazil; Escola Superior de Agricultura Luiz de Queiroz, Universidade de São Paulo (LMUSP), Brazil; Instituto de Desenvolvimento Sustentável Mamirauá (IDSM), Brazil; Instituto Nacional de Pesquisas da Amazônia (INPA), Brazil; MZUFV; Royal Ontario Museum (ROM), Canada; UFMT; and Universidade Federal da Paraíba (UFPB). Specimen sampling covers most of the distribution of the genus (Fig. 1).

Figure 1 Occurrence localities of the long-legged bat (Macrophyllum macrophyllum).

Literature records of Macrophyllum macrophyllum and collecting localities of the sequenced specimens. For locality information see Table S1.

Genomic DNA was extracted from 21 specimens from Brazil according to the protocol described by Aljanabi & Martinez (1997), with only slight modifications. Polymerase chain reactions (PCR) used MVZ05 and MVZ16 primers (Smith & Patton, 1993) for amplification of approximately 800 basepairs (bp) of the mitochondrial Cytochrome b gene (cytb), following the protocol of Saldanha et al. (2019). The PCR purification, preparation and sequencing were outsourced to the “Biotecnologia, Pesquisa e Inovação - BPI, São Paulo, Brazil”. Sequences of both forward and reverse directions were aligned to assemble the consensus in the software Geneious v. 7.1.3 (Kearse et al., 2012) and deposited in GenBank (Table 1). Additionally, 11 specimens at ROM from Ecuador, Guyana, Panama, and Suriname were sequenced for 1,140 bp of cytb following the protocol of Lim et al. (2008). DNA extraction was done using a phenol-chloroform procedure, PCR amplification used primers LGL765 and LGL766 Bickham, Wood & Patton (1995), and nucleotide sequencing was performed on an Applied Biosystems 3730 analyzer in the Laboratory of Molecular Systematics at ROM. Sequences were verified and aligned using Sequencher v. 4.8 (Gene Codes Corporation, Ann Arbor, MI, USA).

Phylogenetic analyses

All sequences were aligned using the MUSCLE algorithm’s default settings (Edgar, 2004). The final character matrix used in the phylogenetic analyses comprised 40 taxa. In addition to sequences from 33 individuals of Macrophyllum (Table 1), we included seven sequences of the following closely related species of phyllostomid bats (GenBank accession numbers in parentheses): Trachops cirrhosus (FJ155483, MH102398, MH102399), Lophostoma silvicola (JF923851), Phyllostomus hastatus (FJ155479), Chrotopterus auritus (FJ155481), and Vampyrum spectrum (FJ155482).

Phylogenetic trees for the cytb gene were inferred using maximum likelihood (ML) and Bayesian inference (BI) approaches. The ML analysis done in IQ-TREE version 1.6.12 (Nguyen et al., 2015), with ultrafast bootstrap values calculated on the consensus tree based on 1,000 replicates (Hoang et al., 2017). The BI analysis was conducted in MrBayes 3.2.7a via the online CIPRES platform (Miller, Pfeiffer & Schwartz, 2010; Ronquist et al., 2012). Four Markov Chain Monte Carlo (MCMC) chains were run for 40,000,000 generations, sampling every 4,000 generations. A burn-in of 30% was discarded. Convergence was verified using Tracer 1.7.1 (Rambaut et al., 2018), by evaluating effective sample sizes (ESS > 200) and trace plots to ensure proper mixing and stationarity of MCMC chains. Support was assessed using posterior probabilities. In both BI and ML analyses, we divided our data into three partitions (1st, 2nd, and 3rd codon positions) and estimated the best substitution scheme using IQ-TREE. We used the “-mset mrbayes” command in IQ-TREE, so the model chosen would be appliable to MrBayes. The chosen models for each partition were SYM+G4 (1st codon), HKY+F+G4 (2nd codon), GTR+F+I+G4 (3rd codon).

Population analyses

We examined genetic variation in the sample of 33 specimens of M. macrophyllum. Due to the missing data in some shorter sequences, we trimmed them to 801 base pairs for the population analyses. To verify if there is geographic structuring among the main haplogroups, we divided our samples using three distinct approaches (Table 1). One geographic approach considered hydrographic basins, following the classification proposed in the HydroSheds database (Lehner & Grill, 2013), and resulted in the following five groups: Amazon, La Plata, Northeast South America, Pacific, and São Francisco. In the second approach, individuals were grouped according to the ecosystem predominant for the locality, resulting in five groups: Amazonia, Central American forests, Cerrado, Guianas Shield, and Pantanal. For the third approach, we used a Bayesian hierarchical clustering (BHC) analysis to determine the most likely number of genetic clusters (K) in our dataset (Tonkin-Hill et al., 2019). The BHC analysis was done using the “fastbaps” package in R, using k.init = 10 to obtain the best partition under a Dirichlet Process Mixture model (R Core Team, 2020). Four populations were defined in this method (Table 1), as follows: Pop1: Brazil (Cuiabá, Mato Grosso); Pop2: Brazil (Minas Gerais, Pará, Santo Antonio do Leverger. Mato Grosso), Guyana, Suriname, and Venezuela; Pop3: Brazil (Amazonas), Ecuador (Napo); and Pop4: Panama.

Molecular variance (AMOVA) was calculated to evaluate the distribution of genetic variability within and among the defined populations and groups, based on the three proposed approaches: hydrographic basins, ecosystems, and BHC. Additionally, overall and pairwise fixation indices (Fst) were computed for each of the three approaches, with statistical significance (P- values) determined through 1,000 permutations. Both AMOVA and Fst were performed using Arlequin 3.5 (Excoffier & Lischer, 2010). Pairwise sequence divergence among the populations recognized in the BHC were estimated with a Kimura 2-parameter model using the dist.dna function in “ape” R package (Paradis & Schliep, 2019).

A minimum spanning haplotype network was constructed using PopArt v.17, with epsilon set to zero (Bandelt, Forster & Röhl, 1999; Leigh & Bryant, 2015). To verify correlation between pairwise genetic distances and geographic distances between populations we performed a Mantel test (10,000 permutations) in R using the “ape”, “geodist” (Padgham & Sumner, 2021) and “vegan” packages (Oksanen et al., 2022).

Results

Natural history

The literature review, together with our new data, yielded 73 publications and 134 records of the species that account for 32 observations of day roosts used by M. macrophyllum (Table S1). We divided the roosts into eight types (Table 2); most of the roosts (43.75%) were in drainage culverts, followed by caves (15.63%). Also, species of the genera Carollia and Glossophaga were typically found roosting with M. macrophyllum. Groups usually were composed of 3-8 individuals (Hill & Bown, 1963; Peracchi & Albuquerque, 1971; Simmons & Voss, 1998; Tavares & Anciães, 1998; Stutz et al., 2004; Voss et al., 2016) but larger colonies of approximately 50 were also recorded (Seymour & Dickerman, 1982; Peracchi, Raimundo & Tannure, 1984).

Table 2 Roosts used by Macrophyllum macrophyllum and other species cohabiting them.a

	Abandoned buildings/ruins	Cave	Concrete bridge gap	Crevices in rock	Culvert/water tunnel	On dead tree trunk	Under stilt house	Wooden deck	
Number of records	3	5	1	2	14	3	3	1	
Species cohabiting roost									
Carollia sp.	1	1	1	–	9	–	–	–	
Desmodus rotundus	–	1	–	–	–	–	–	–	
Glossophaga sp.	1	–	–	–	5	–	–	–	
Myotis albescens	–	–	–	–	–	–	1	–	
Pteronotus mesoamericanus	–	–	–	–	1	–	–	–	
Rhynchonycteris naso	–	–	1	–	–	–	2	–	
Trachops cirrhosus	–	–	–	–	1	–	–	–	
Notes.

a Records were based on Goldman (1920), Thomas (1928), Ruschi (1953), Lay (1962), Hill & Bown (1963), Handley (1957), Handley (1966), Linares (1966), Starrett & Casebeer (1968), Fornes, Delpietro & Massoia (1969), Peracchi & Albuquerque (1971), Harrison & Pendleton, (1974), Taddei (1975), Handley (1976), Reis & Schubart (1979), Seymour & Dickerman (1982), Coimbra et al., 1982), Peracchi, Raimundo & Tannure (1984), Marques (1985), Patterson (1992), Simmons & Voss (1998), Tavares & Anciães (1998), Stutz et al. (2004), Faria, Soares-Santos & Sampaio (2006), Voss et al. (2016) and present study.

Of the 22 specimens obtained in Mato Grosso state, 14 were from a colony of approximately 30 recorded under a wooden fishing deck (WFD) on Rio São Lourenço during the rainy season in February 10th and 11th, 2021. The bats were hanging below the platform, a few centimeters above the water (Figs. 2A, 2B). Of these 14 specimens, 6 were males and 8 were pregnant females. Five specimens were sampled in a daytime roost in an expansion gap of a concrete bridge (GCB) on highway MT-040, crossing Rio Cuiabá Mirim, on October 20th, 2022 (Figs. 2C, 2D). Three specimens were caught in a drainage culvert made of concrete (DCC), on December 5th, 2022 (one specimen), and on December 26th, 2023 (two specimens). The culvert passed under a dirt road, and through it flowed a small creek (Figs. 2E, 2F). In both GCB and DCC, only males were captured. Seven of the eight females in the WFD roost were pregnant. Cohabitation was observed in two of the three roosts, with the species Rhynchonycteris naso and Carollia perspicillata recorded in GCB, and C. perspicillata and Glossophaga soricina in DCC (Table 2).

Figure 2 Daytime roosts of Macrophyllum macrophyllum.

(A) Wooden fishing deck on Rio São Lourenço. (B) Underside of fishing deck. (C) Concrete bridge on highway MT-040, over Rio Cuiabá-Mirim. (D) Expansion gap of the concrete bridge, where the colony was located. (E) Bridge on dirt road, crossing a small creek along a riparian forest. (F) Culvert of the bridge, inside which the colony was found. All localities are in the Brazilian state of Mato Grosso (see Table 1). Photographs by Thiago B.F. Semedo.

Phylogenetics

Mitochondrial DNA sequences of the cytb gene confirmed the monophyly of Macrophyllum relative to the outgroup taxa in other genera of phyllostomid bats, with maximum clade support values both in the BI and ML analyses. The BI phylogeny recovered five highly supported lineages within Macrophyllum, with posterior probability values ranging from 0.92 to 1 (Fig. 3A), although many of the more basal relationships were not well resolved. Specimens from Panamá were well supported (PP = 1) and sister to a clade from the Amazonian lowlands that had low support (PP = 0.38). Part of the western lowland Amazonian clade from Brazil is highly supported (PP = 0.92), as well as the eastern Ecuadorian clades (PP = 1). The clade including specimens from lowland Amazonia and Panamá is the sister group to a poorly supported (PP = 0.47) clade that consisted of all other South American samples including specimens from Cuiabá in the western Cerrado in a well-supported clade (PP = 1) sister to all remaining clades including the Brazilian Pantanal, Brazilian eastern Cerrado (Jequitaí), and the Guiana Shield. The Pantanal clade (specimens from Baia São João and Mimoso, Mato Grosso) was highly supported (PP = 0.96), as well as the Guianan clade (specimens from Guyana, Suriname, and Venezuela; PP = 0.99).

Figure 3 Phylogeny of Macrophyllum macrophyllum, as inferred from the cytb data.

(A) Tree inferred in MrBayes 3.2.7a. Numbers above branches are Bayesian posterior probabilities. (B) Tree inferred in IQ-TREE version 1.6.12. Number above branches are the bootstrap values. Colours correspond to the haplotype clusters identified via Bayesian hierarchical clustering (see Fig. 4, Table 1 and Materials & Methods). The five geographic lineages discussed in the text are indicated in the central column.

The ML-based phylogenetic inference tree had UltraFast bootstrap values between 63 and 100, but some of the more terminal relationships are unresolved (Fig. 3B). As in the BI tree, the Cuiabá clade had high support (BS = 100). The clade containing specimens from Pantanal was also highly supported (BS = 97), as well as the Guiana Shield clade (BS = 97). The western Amazonian clade, containing specimens from Ecuador and Amazonas, Brazil had lower support (BS = 88). Topology did not differ substantially between BI and ML, the primary difference being the position of the Cuiabá clade from western Cerrado, which is sister to the remaining specimens in the ML tree, whereas in the BI tree it is sister to a clade containing the eastern Cerrado, Pantanal, and Guianas (Fig. 3).

Genetic diversity and population structure

We identified 22 haplotypes from the aligned 801 base pair fragment of the cytb gene (Fig. 4). The haplotype network had a “reciprocally monophyletic” structure, in which multiple lineages are each connected by a long branch characterized by numerous mutations (Jenkins, Castilho & Stevens, 2018). No haplotypes were shared between different localities and the Mantel test indicated a significant correlation between geographic distance and genetic variation in the samples (R = 0.144; p = 0.047).

Figure 4 Geographic distribution of the genetic lineages and haplotype network of Macrophyllum macrophyllum.

(A) Geographic distribution of the haplogroups defined via Bayesian hierarchical clustering. Colors in the map represent hydrographic basins, and dashed-filled areas indicate the ecosystems mentioned. Solid lines represent country boundaries, while dashed lines indicate Brazilian federal states from which sequences were obtained. Country names are in regular fonts and federal states are in italics. (B) Minimum spanning haplotype network was constructed using PopArt v.17, with epsilon set to zero. Locality number in the network correspond to the map localities.

The geographic approach that considered four populations based on BHC had the highest Fst values, followed by the distinct ecosystems approach and the hydrographic basins had the lowest Fst (Table 3). In the BHC, the largest Fst values were observed between Pop1 (Cuiabá) and the other three populations (Fig. 5). The largest (but not statistically significant) difference was between Pop1 and Pop4 (Panama). Considering the distinct ecosystems, the largest Fst values were between Central America (Panama) and Pantanal specimens (Fig. 5). In the geographic approach that classified the populations into five hydrographic basins, the highest Fst values were observed between specimens from the Pacific coast (Panama) and São Francisco (Minas Gerais). Pair-wise genetic distances varied from 0.03, between Pop2 x Pop3 and Pop2 x Pop4, to 0.06, between Pop1 x Pop3 and Pop1 x Pop4 (Table 4).

Table 3 Analysis of molecular variance (AMOVA) between populations of Macrophyllum macrophyllum.

Analysis of molecular variance (AMOVA) between populations of Macrophyllum macrophyllum based on 801 loci of the cytochrome b gene calculated in Arlequin v. 3.5. (A) Values calculated based on four populations defined by Bayesian hierarchical clustering. (B) Values calculated based on five populations defined by ecosystems of occurrence. (C) Values calculated based on hydrographic basin the specimens were collected.

Locus by locus AMOVA	Sum of squares	Percentage of variation	FST	P value*	
(A) Source of variation (BHC)					
Among populations	387.736	68.065	0.681	<0.001	
Within populations	306.743	31.935			
Total	694.479				
(B) Source of variation (ecosystem)					
Among populations	436.009	66.474	0.665	<0.001	
Within populations	258.471	33.526			
Total	694.480				
(C) Source of variation (hydrographic basin)					
Among populations	333.589	53.008	0.530	<0.001	
Within populations	360.890	46.992			
Total	694.479				
Notes.

* Significance based on 1,000 permutations.

Figure 5 FST values among populations of Macrophyllum macrophyllum.

Populations were defined by Bayesian hierarchical clustering (BHC), ecosystems of occurrence, and hydrographic basin.

Table 4 Pair-wise cytochrome b sequence divergence among the four populations of Macrophyllum macrophyllum defined here.

Cytochrome b sequence divergence (mean ± standard deviation) among the four populations of Macrophyllum macrophyllum defined by Bayesian hierarchical clustering, estimated using the Kimura 2-parameter model.

	Pop1	Pop2	Pop3	Pop4	
Pop1	0				
Pop2	0.05 ± 0.01	0.01 ± 0.01			
Pop3	0.06 ± 0.02	0.03 ± 0.02	0.02 ± 0.02		
Pop4	0.06 ± 0.02	0.03 ± 0.02	0.04 ± 0.02	0.001 ± 0.02	

Discussion

The cytb sequences reported in this study represent the most extensive molecular dataset assembled for Macrophyllum macrophyllum, covering most of the species’ geographic distribution. Phylogenetic analyses based on this dataset revealed geographic structuring within Macrophyllum. Our findings support a previous study that utilized DNA barcoding sequences (∼657 bp) of the Cytochrome c oxidase subunit I (COI) gene and identified a sister-group relationship between samples from Panama and Amazonia, as well as a distinct clade from the Guiana Shield, but no samples from Brazil were used (Clare et al., 2011).

Our initial hypothesis that hydrographic basins would be important to explain geographic structuring in Macrophyllum was not supported by the observed genetic pattern. Instead, the defined clusters are more related to the ecosystem regions. Interestingly, localities that are relatively close geographically, but are in distinct ecosystems, particularly those in the Pantanal (specimens from Santo Antonio do Leverger) and the locality in the Cerrado (Cuiabá) that are about 150 km apart, were distinct genetically (Figs. 4, 5; Table 4).

Some of the clusters identified here correspond to known biogeographic units for other vertebrates, such as the Guiana Shield (Lim & Tavares, 2012). This region has long been recognized as a subdivision of Amazonia which has been relatively stable during the Cenozoic (Hoorn et al., 2010) and supporting a high diversity of endemic vertebrates (Vacher et al., 2024). In our genetic cluster analysis, specimens from the Guiana Shield grouped with those from southeastern and central Brazil, as well as eastern Amazonia. However, in the phylogenetic analysis, the Guiana Shield formed a highly supported clade (Figs. 3, 4). These findings suggest that incorporating underrepresented populations and additional genetic markers, such as loci from the Y chromosome and autosomes, may refine the haplotype structure observed here.

Conversely, specimens from central and western Amazonia formed a distinct cluster. In the phylogenetic analysis, eastern Amazonia was weakly supported in a clade with other Amazonian samples (Fig. 3). In the haplotype network, the eastern Amazonian sample (UFPB PR20207) from Rio Xingu exhibited 14 mutational differences from central Amazonia but only 11 from the Pantanal, influencing its assignment to the genetic cluster Pop2 (Fig. 4). Given this pattern, UFPB PR20207 may represent an admixed population with genetic contributions from both Pop2 and Pop3, highlighting potential historical gene flow between these groups.

The finding that specimens from Central America (Pop4 in Fig. 5) constitute a distinct cluster corroborates several previous studies that indicate the Andes as an important geographic barrier for phyllostomid bats (Koopman, 1978; Larsen et al., 2007; Velazco & Patterson, 2008; Lim, Loureiro & Garbino, 2020; Esquivel et al., 2022). However, this population was not so distinct genetically from the central and western Amazonian populations (Pop3 in Fig. 5), which may suggest a more recent trans-Andean expansion as indicated by their close but poorly supported relationship in the phylogenetic tree (Fig. 3). We recommend further sampling from Central America and Mexico to refine the phylogenetic positioning and haplotypic diversity of trans-Andean populations.

When comparing the phylogeographic patterns of other phyllostomid bats, no consistent trend emerges. For instance, there is strong phylogenetic structuring in the small frugivorous species Rhinophylla pumilio (Silva et al., 2024), and in the vampire bat Desmodus rotundus (Martins et al., 2007). In contrast, larger frugivores, such as Artibeus planirostris, Chiroderma doriae, and Chiroderma villosum, lack geographically structured haplotypes (Larsen et al., 2007; Garbino, Lim & Tavares, 2020). On the other hand, species such as Carollia perspicillata and Trachops cirrhosus exhibit a mix of geographically restricted clades alongside widely dispersed haplogroups (Pavan et al., 2011; Camacho et al., 2024). Considering these studies, M. macrophyllum has a pattern more similar to R. pumilio and D. rotundus.

However, we emphasize that while cytochrome b is useful for identifying distinct species (Baker & Bradley, 2006), it reflects only the mitochondrial genome, which may differ from nuclear patterns (Lebedev et al., 2021; Klicka et al., 2024). Therefore, incorporating DNA data from additional loci will be essential for clarifying species-level divergences within Macrophyllum.

We identified eight different types of roosts used by M. macrophyllum, two of which are reported for the first time in this study. Our findings support previous assumptions that the species shelters in dark vaulted cavities and commonly roosts alongside members of the genera Carollia and Glossophaga (Goldman, 1920; Ruschi, 1953; Lay, 1962; Fornes, Delpietro & Massoia, 1969; Harrison, 1975; Reis & Schubart, 1979; Coimbra et al., 1982; Tavares & Anciães, 1998). Additionally, our results underscore the significance of water culverts as artificial roosting sites for this species (Table 2). We recommend that future field studies carefully examine Carollia and Glossophaga colonies in culverts, as these roosts may harbor M. macrophyllum that are easily overlooked.

Conclusions

This study represents the first investigation into the genetic structure of Macrophyllum macrophyllum across most of its distributional range. Our findings reveal evidence of geographic structuring within the species, potentially indicating diverging lineages. However, further integrative studies, particularly with expanded sampling in Central America and southeastern Brazil and incorporating nuclear data, are required to assess whether taxonomic reclassification is warranted. Observations from natural history emphasize the significance of culverts as artificial roosts for this species and the importance of active searches to sample this species. Overall, our results offer a clearer understanding of the genetic variability within M. macrophyllum, underscoring the need for a more comprehensive examination of its taxonomy and morphological and genetic diversity.

Supplemental Information

Supplemental Information 1 Localities of Macrophyllum macrophyllum gleaned from the literature and used to produce the map in Fig. 1

Supplemental Information 2 Bayesian phylogeneyic inference analysis run in MrBayes (Nexus)

Supplemental Information 3 Bayesian inference cytochrome b phylogeny of Macrophyllum macrophyllum (Newick)

Supplemental Information 4 Maximum likelihood-based phylogeny of the cytochrome b gene of Macrophyllum macrophyllum (Newick)

Supplemental Information 5 Aligned cytochrome b DNA sequences (FASTA)

Two anonymous reviewers provided valuable insights and recommendations. We are most grateful to Alexandre Percequillo, Fernando A. Perini, Fred Victor Oliveira, Gerson Paulino Lopes, Patrício Rocha, and Tamily Carvalho Melo Santos for providing tissue samples of recently collected specimens. Cayo A.R. Dias provided advice on the phylogeographic analyses. Marcelo Nogueira and Marlon Zortea gave the precise coordinates for specimens from Acre and Goiás, respectively.

Additional Information and Declarations

Competing Interests

Author Contributions

Animal Ethics

DNA Deposition

Data Availability

The authors declare there are no competing interests.

Guilherme S.T. Garbino conceived and designed the experiments, performed the experiments, analyzed the data, prepared figures and/or tables, authored or reviewed drafts of the article, and approved the final draft.

Thiago Borges Fernandes Semedo conceived and designed the experiments, analyzed the data, prepared figures and/or tables, authored or reviewed drafts of the article, and approved the final draft.

Juliane Saldanha conceived and designed the experiments, performed the experiments, analyzed the data, authored or reviewed drafts of the article, and approved the final draft.

Daniela Cristina Ferreira conceived and designed the experiments, performed the experiments, analyzed the data, authored or reviewed drafts of the article, and approved the final draft.

Rogerio Vieira Rossi analyzed the data, authored or reviewed drafts of the article, provided critical information on the study subject, and approved the final draft.

Maria Nazareth Ferreira da Silva analyzed the data, authored or reviewed drafts of the article, and approved the final draft.

Burton K. Lim conceived and designed the experiments, performed the experiments, analyzed the data, authored or reviewed drafts of the article, and approved the final draft.

The following information was supplied relating to ethical approvals (i.e., approving body and any reference numbers):

Comissão de Ética no Uso de Animais da Universidade Federal de Viçosa (Ceua-UFV) (https://www.ceua.ufv.br/).

The following information was supplied regarding the deposition of DNA sequences:

The cytochrome b sequences are available in the Supplemental Files and at GenBank and Table 1: PV009324 to PV009334.

The following information was supplied regarding data availability:

The phylogenetic trees and the phylogenetic analyses are available in the Supplemental Files.

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
