# Peer review of "Phylogeographic analysis of long-legged bats, Macrophyllum macrophyllum, with notes on roosting behavior and natural history"

_PeerJ, doi:10.7717/peerj.19432_

## Round 0.1 · original submission · Minor Revisions

Thank you for your submission to PeerJ.

Please change this version as reviewers' comments.

Reviewer 1 ·

Basic reporting

No comment.

Experimental design

Lines 164-166: it's unclear how the convergence of MCMC chains were checked.
Line 184: please provide the parameters (eg k.init) used in the fastbaps analyses.

Validity of the findings

My main concern is about the genetic population assignments. In the abstract (lines 39-40), the authors mentioned 5 phylogeographic lineages based on the phylogenetic tree, but these lineages were not visually labeled in Figure 3 nor included in Table 1. In addition, the sample UFPB from Brazil Para is labeled as pop3 (green) in Figure 3 but pop2 (yellow) in Figure 4 and Table 1. Please revise this and clarify the phylogeographic population assignments. Given the positions of UFPB in the phylogenetic tree, the haplotype network, and the geographic location, it looks like this sample may represent an admixed population between pop2 and pop3? Assigning it to either of the two populations may lead to higher within-population genetic distance (pop2xpop2, pop3xpop3) and lower between-population (pop2xpop3) genetic distance, which is exactly what is shown in Table 3. I think this should be discussed. On the other hand, sites 2-6 and 14-16 form separate clusters geographically, in the phylogenetic tree, and in the haplotype network. I wonder how much BHC supported the clustering of these sites into one genetic population (pop2). What was the k.init parameter used to run fastbaps? Since the k.init paramter should be “significantly larger than the number of clusters you expect” (https://github.com/gtonkinhill/fastbaps), maybe using a larger k.init could help get a more reasonable result. In the minimum, I think these edge cases (UFPB and sites 2-6) deserve more discussion.

Below are my minor comments.

Lines 179-181: I suggest also illustrating these ecosystem regions on the map figure (e.g., add a labeled polygon for each ecosystem region).

The maps in Figure 1 and Figure 4A can be combined. “Hydrographic basins” should also be mentioned in the caption. I suggest adding text labels of geographic sites on the map. The sampling sites were referred to by their locality names throughout the text, which may be difficult to follow without a map. Maybe label the country on the map and add location names or abbreviations next to the sampling sites, consistent with the in-text site names and the labels in Figure 3 and Figure 4B.

Lines 227-230: The Cuiaba clade is the major difference between Bayesian and ML trees, but this is not easy to see in Figure 3. I suggest highlighting this branch (e.g., in a different color).

Figure 3: I suggest also labeling the 5 phylogeographic lineages, in addition to the BHC-assigned 4 populations. In addition, the tip names (e.g., Mato Grosso) are not informative in this context. I suggest using names consistent with those in Figure 4B to indicate the exact sampling sites. Lastly, I suggest including the outgroups in the phylogenetic tree to also present the results of the species relationships.

Figure 3 is mentioned before Figure 2. Maybe move the “natural history” section in front of the genetic results? This is also more logical as Figure 2 and the related section shows where these bats roost and were sampled, then one can continue with the genetic analyses of these samples.

Although cytb is a very useful genetic marker, it only captures the mitochondrial structure which can differ from the nuclear pattern. I think the caveats and limitations of using mitochondrial data only should be discussed more clearly.

Additional comments

Overall I think this study improves our understanding of the under-studied long-legged bat populations and diversity. The new cytb sequences of this species are also valuable resources that will improve future studies on Neotropical bats.

Reviewer 2 ·

Basic reporting

The manuscript is well presented, the language is clear and formal. Some cites were included in different type and size of font (line 256, 291, 323).
The paper provides unpublished information on the haplotypic diversity of the species. It also provides information on the natural history of the species. The information presented is relevant and meets the profile of the journal.
On the other hand, I find some areas for consideration in the manuscript. In the introduction they could expand on the general biology of the species, considering that they provide more information in this regard, they could give a little more extended context.

Experimental design

The methodology is adequate to respond to the general objective, although the search for records does not seem to me to be part of the sampling, since it is evident that this is very limited in several of the areas of the distribution of Macrophyllum, and even in the north it is very poor (Central America)
I do not consider it necessary to mention in the methods section the search for information about the species, since we must always document and rely on the specialized literature, and in the results section they do not mention anything about this search or how it influenced the analyses, they do not even mention how many sources of information they found in this search, so I suggest that they eliminate it and include the relevant information from this search in the introduction and discussion, or else that they mention the results of the search and the contribution to the work.

Validity of the findings

Regarding the sample size, it is important to mention how the sample size influences the interpretation of the results. Especially because of the lack of organisms in several parts of the distribution.
About the interpretation of the results, I have a question that I think is important: Why did you consider the Guyana, Suriname, and Venezuela group to be part of the southernmost group? That is, in the genealogies these individuals are observed as a monophyletic clade. In the haplotype network, these samples are also observed separated from the rest of the southern samples, in addition, in the Mantel test you have a statistically significant result, which is interpreted that the greater the geographical distance, there are more genetic variation. Didn't you test a scenario with this group as a different group or maybe one more K?

Additional comments

Introduction.
In this section, you can give more information about the species, such as group or family size, if it is associated with some vegetation type throughout its distribution, i.e. a bit more biological and ecological context.

Line 64. "the largest and endemic "family ...".

Line 80. It is important to note that their work is limited to a specific area of the total Macrophyllum distribution. Since it is a practically Pan-American bat and the samples are only representative of some sites.

Line 83. I recommend that they try to clarify their hypothesis regarding the distribution of bats, since the data and results they present in this paper do not describe the total distribution of Macrophyllum. Do you expect similar populatin genetic structure in the northern of distribution?

Materials & Methods
Specimen sampling and literature review.
line 90 It is not entirely clear how the samples were collected outside Brazil. Was the same method used at the sites in Ecuador, Guyana, Suriname, Panama and Venezuela?

Results
You have used a couple of sister species, but they are not included in the phylogenies, why is that?

Discussion
Line 276. I am not sure if most of the distribution of the species is covered, there are large areas and hydrographic basins used as reference in this work that have no samples.
How limited sampling affects the analysis of results. That is, there are no samples in Central America or North America, but in South America, most of the areas where there are records of the species' presence are not represented. How this affects the genetic structure of the species.
Line 283. You mention that the structure is more related to the type of ecosystem than to the hydrographic basins. However, the map does not show the types of ecosystems and does not visualize this relationship, it only shows the hydrographic basins.

---

## Round 0.2 · accepted · Accept

Congratulations!

Thank you for your submission to PeerJ.

Reviewer 1 ·

Basic reporting

I enjoyed reading the revised manuscript. It is clear and well presented. My major concerns have been addressed.

Experimental design

Lines 171-173: Please specify the threshold of ESS you used (e.g., normally ESS>200 is considered as a good sign for a MCMC parameter estimation). Besides, this sentence is a repeat: “Support was evaluated based on posterior probabilities.”.

Validity of the findings

Lines 272-274: When re-reading this section, I find myself wondering what is the main goal of estimating the Fst? Because the BHC clustering is based on genetics while the hydrographic basins and ecosystem methods are based on ecology, it is totally expected that the BHC clusters should have the highest pairwise Fst. Therefore, it would be a reasoning loop if Fst was used to validate that BHC was the best clustering method. I don't think this was the reasoning, but maybe it would be better to add one sentence of clarification.

Additional comments

Line 308: Y-DNA sounds a bit strange. Maybe “loci from the Y chromoome and autosomes”?

Reviewer 2 ·

Basic reporting

Manuscript #113702 meets the journal's publication criteria. The authors have responded clearly and adequately to the suggestions and questions I raised in my review. I congratulate the authors as their work contributes to an understudied species, and I hope (and if they have the opportunity) that further studies can be conducted to expand the sample in number and geographic coverage.

Experimental design

The methods and analyses presented in the paper are congruent and adequate to meet the objectives of the manuscript. The revisions provided clarity on the methods and objectively justified the choices made with respect to the analyses applied.

Validity of the findings

In this new version, they clarified and justified the interpretations of the results based on the methods. They included very valuable information from previous bibliographic records and highlighted the efforts made to obtain information on the natural history of the species.

Additional comments

I recommend that you double-check the size and font of some of the quotes, as they are not consistent, e.g: Lines 118, 219-22, 301, 346-348.